# Differential Neuropathology, Genetics, and Transcriptomics in Two Kindred Cases with Alzheimer’s Disease and Lewy Body Dementia

**DOI:** 10.3390/biomedicines10071687

**Published:** 2022-07-13

**Authors:** Ilaria Palmieri, Tino Emanuele Poloni, Valentina Medici, Susanna Zucca, Annalisa Davin, Orietta Pansarasa, Mauro Ceroni, Livio Tronconi, Antonio Guaita, Stella Gagliardi, Cristina Cereda

**Affiliations:** 1IRCCS Mondino Foundation, 27100 Pavia, Italy; ilaria.palmieri@mondino.it (I.P.); mauro.ceroni@unipv.it (M.C.); stella.gagliardi@mondino.it (S.G.); cristina.cereda@asst-fbf-sacco.it (C.C.); 2Department of Neurology-Neuropathology and Abbiategrasso Brain Bank, Golgi-Cenci Foundation, Abbiategrasso, 20081 Milan, Italy; e.poloni@golgicenci.it (T.E.P.); v.medici@golgicenci.it (V.M.); a.guaita@golgicenci.it (A.G.); 3Department of Rehabilitation, ASP Golgi-Redaelli, Abbiategrasso, 20081 Milan, Italy; 4enGenome, 27100 Pavia, Italy; szucca@engenome.com; 5Laboratory of Neurobiology and Neurogenetics, Golgi Cenci Foundation, Abbiategrasso, 20081 Milan, Italy; a.davin@golgicenci.it; 6U.O. Medicina Legale, IRCCS Mondino Foundation, 27100 Pavia, Italy; livio.tronconi@mondino.it; 7Unit of Legal Medicine and Forensic Sciences “A. Fornari”, Department of Public Health, Experimental and Forensic Medicine, University of Pavia, 27100 Pavia, Italy; 8Department of Women, Mothers and Neonatal Care, Children’s Hospital “V. Buzzi”, 20100 Milan, Italy

**Keywords:** Alzheimer’s disease, Lewy body dementia, neuropathology, transcriptomics, *substantia nigra*, *hippocampus*, parietal lobe, basal ganglia

## Abstract

Alzheimer’s disease (AD) and Lewy body dementia (LBD) are two different forms of dementia, but their pathology may involve the same cortical areas with overlapping cognitive manifestations. Nonetheless, the clinical phenotype is different due to the topography of the lesions driven by the different underlying molecular processes that arise apart from genetics, causing diverse neurodegeneration. Here, we define the commonalities and differences in the pathological processes of dementia in two kindred cases, a mother and a son, who developed classical AD and an aggressive form of AD/LBD, respectively, through a neuropathological, genetic (next-generation sequencing), and transcriptomic (RNA-seq) comparison of four different brain areas. A genetic analysis did not reveal any pathogenic variants in the principal AD/LBD-causative genes. RNA sequencing highlighted high transcriptional dysregulation within the substantia nigra in the AD/LBD case, while the AD case showed lower transcriptional dysregulation, with the parietal lobe being the most involved brain area. The hippocampus (the most degenerated area) and basal ganglia (lacking specific lesions) expressed the lowest level of dysregulation. Our data suggest that there is a link between transcriptional dysregulation and the amount of tissue damage accumulated across time, assessed through neuropathology. Moreover, we highlight that the molecular bases of AD and LBD follow very different pathways, which underlie their neuropathological signatures. Indeed, the transcriptome profiling through RNA sequencing may be an important tool in flanking the neuropathological analysis for a deeper understanding of AD and LBD pathogenesis.

## 1. Introduction

Alzheimer’s disease (AD) represents the major type of dementia worldwide [1], characterized by progressive cognitive decline with no motor function impairment until the late stages of the disease, due to the late involvement of basal ganglia (BG). The AD clinical course reflects the progressive spread of lesions, typically from the entorhinal cortex and hippocampus (HiC) to the parietal lobe (PL) and subsequently to the whole brain. Extracellular accumulation of amyloid-beta (Aβ) peptides in senile plaques and the intraneural aggregation of hyperphosphorylated tau protein (pTAU) forming neurofibrillary tangles (NFTs) and threads, and neuritic plaques (NPs) are the main neuropathological hallmarks of the disease. Accordingly, the neuropathological diagnosis of AD is based on the combination of the Aβ plaque score, the Braak NFT stages, and the CERAD (Consortium to Establish a Registry for AD) NP score, which together constitute the ABC criteria for the diagnosis of AD-related pathologies [2,3,4,5]. Lewy body dementia (LBD) is considered the second most common cause of degenerative dementia, responsible for about 20% of cases [6,7,8,9]. LBD may involve the same cortical areas of AD with overlapping clinical manifestations. In addition, LBD presents clinical peculiarities distinguishing it from AD, due to the early involvement of the limbic system and brainstem, including fluctuations of mental state, psychosis and hallucinations, REM behavior disorder, and parkinsonism. The motor function may be precociously impaired due to the early involvement of the midbrain, particularly of the substantia nigra (SN). Indeed, SN pathology is a typical feature of parkinsonian syndromes, such as Parkinson’s disease (PD) and PD dementia (PDD) [10]. The principal hallmark of LBD, PD, and PDD is the accumulation of α-synuclein (α-syn) in cortical and brainstem neurons into aggregates named Lewy bodies (LBs) and Lewy neurites (LNs) [11]. LBDs and LNs constitute the so-called Lewy-type synucleinopathy (LTS), and their initial topography in the olfactory bulb, brainstem, limbic system, and/or neocortex and their subsequent spreading determine the evolution of clinical pictures belonging to the PDD and LBD spectrum [12].

It has long been known that AD and LBD pathologies are often coexisting [13], and AD/LBD cases are increasingly recognized and differentially contribute to the clinical picture of dementia [10]. AD and LBD share pathological aspects, such as the co-occurrence of different proteinopathies, including Aβ, tau, and α-syn [14]. More recently, neuronal deposits of another protein involved in neurodegeneration, the transactive response DNA-binding protein (TDP-43), have been found in the amygdala, HiC, and frontal cortex of aging brains. TDP-43 deposits in the limbic structures are now identified with the LATE acronym (limbic-predominant age-related TDP-43 encephalopathy) and are frequently associated with hippocampal sclerosis and combined with the other proteinopathies [15]. This copresence of multiple proteinopathies causes interactions at the molecular level with a possible synergistic detrimental effect on synaptic and neuronal function and additive effects on brain pathology [16,17]. Amyloid deposition is a pathological hallmark not only in AD but also in LBD, differentiating LBD from PD and indicating that abundant cortical Aβ deposits promote diverse neurodegenerative processes facilitating the onset of dementia [18]. Different is the case of dementias due to frontotemporal lobe degeneration associated with TAU pathology (FTLD-TAU) or TDP-43 pathology. FTLD-TAU shares TAU pathology with AD but shows different topography and histological pictures with no or scarce amyloid deposition in the brain cortex, also due to the age of onset that is generally earlier in FTLD than in AD [19]. Nonetheless, the distinction between pathologies is not absolute, as it occurs in the frontal variant of AD [20,21], and in AD associated with LATE. Especially in old age, the onset of “nonpure” clinical pictures and frequent copathologies make the etiological definition of dementias very complex. This is why, despite the identification of increasingly precise biomarkers, neuropathological characterization is needed to define the diagnosis.

AD and LBD also share similar genetics since variants in common genes have been described to date [22,23,24,25,26]. In addition to the three major genes associated with AD, such as *APP*, *PSEN1,* and *PSEN2*, thanks to large GWAS and meta-analysis, up to 36 loci have been reported as possible contributors to AD [27]. Among these, the Ԑ4 allele of the *APOE* gene remains the strongest genetic risk factor for AD [28], increasing the AD risk of ∼2–5-fold in heterozygous carriers and up to 15-fold in homozygous carriers. Fewer genetic loci have been instead linked to LBD, and the ones that were confirmed by independent GWAS studies are *GBA*, *SNCA,* and *APOE* [29,30]. Only recently, new candidate loci, such as *CNTN1*, *BCL7C*/*STX1B,* and *SCARB2,* have been linked to LBD [27]. Through RNA sequencing (RNA-seq), transcriptional profiling of LBD and AD cases has been performed; however, limited information on the transcriptomes of the SN and the PL, which are key regions involved in LBD and AD neuropathology, is present in the literature [31,32,33,34]. Moreover, familial cases of dementia with common genetic bases may display a very different clinical phenotype, underlying deep transcriptional differences that arise apart from genetics [35]. In this paper, we dealt with two kindred cases, a mother and a son, who developed two different forms of dementia, classical AD and an aggressive form of AD/LBD dementia, respectively. To define commonalities and differences in the pathological processes of dementia, we addressed the two clinical cases and a control case with different approaches, such as the neuropathological, genetic, and transcriptomic profiling of four brain regions. We point out the great differential gene expression in LBD compared to AD, in terms of different transcriptomic profiles.

## 2. Materials and Methods

### 2.1. Clinical Assessment

All clinical assessments were performed by the same team (neurologists, geriatricians, and neuropsychologists), according to the Abbiategrasso Brain Bank protocol (ABB) [36]. The ABB autopsy and sampling protocol were approved by the Ethics Committee of the University of Pavia on 6 October 2009 (Committee report 3/2009). The study procedures were in accordance with the principles outlined in the Declaration of Helsinki of 1964 and the following amendments. The three patients taken under investigation were the 95-year-old mother (BB105), her 72-year-old son (BB181), and a nondemented control individual, a 79-year-old man, who died of liver cancer without any cognitive impairment (BB118). The choice of the control relied on (1) the age at death, to minimize biases in relation to RNA-seq analysis, which is sensitive to age; (2) the neuropathology examination should have been the clearest possible, with no neurodegenerative pathology. All three subjects were of the same ethnic group (Caucasian), and all of them came from the same geographical area. All were followed longitudinally and underwent serial neuromotor and neurocognitive evaluations including global cognition and a complete neuropsychological assessment of specific cognitive domains (memory, attention, executive functions, language, and visuospatial abilities). Clinical dementia rating (CDR) was used for a synthetic definition of dementia severity [37].

### 2.2. Tissue Collection, Preparation, and Immunohistochemistry

Immediately postmortem, the cerebral hemispheres of the three individuals were harvested following the procedure published in Poloni et al. 2020 [36]. Briefly, the hemispheres were directly fresh-cut in slices alternately frozen (at −80 °C) or fixed in 10% phosphate-buffered formalin solution for about 5 days. After, samples were processed for paraffin inclusion and were cut into 8 μm-thick serial sections. Neocortex (frontal, parietal, temporal, and occipital lobes), BG, HiC, cerebellum, and brainstem (SN, pons, medulla oblongata) were analyzed to carry out a detailed neuropathological characterization. Haematoxylin & Eosin (nuclear and cytoplasmic staining), Cresyl Violet (neuronal staining), Luxol Fast Blue (myelin staining), and Gallyas (neuritic plaques staining) were performed to evaluate vascular, architectural, and structural tissue abnormalities. Pretreatments and primary antibodies used for immunohistochemistry are shown in Appendix A. Sections were deparaffinized and pretreated with 3% H_2_O_2_ in PBS for 10 min to neutralize endogenous peroxidase activity; only for some antibodies was a specific additional treatment necessary. To mask nonspecific adsorption sites, a preincubation with 5% normal goat serum was performed for 30 min, and then incubation with the primary antibodies overnight at room temperature followed. The following antibodies were used: 4G8, AT8, α-SYN (KM51), and phosphor-TDP-43 (pS409/410-2) to detect Aβ, pTAU, α-syn, and TDP-43 protein deposits, respectively (Appendix A). On the day after, sections were rinsed in PBS and incubated with the secondary antibody (Envision+ System-HRP labeled Polymer) at a dilution of 1:2 in PBS for 1 h at RT. After several washes in PBS, a chromogen system with diaminobenzidine (Liquid DAB + Substrate Chromogen System) was used to reveal the reaction.

### 2.3. Genetic Analysis

#### 2.3.1. DNA Preparation and Next-Generation Sequencing

DNA was isolated from peripheral blood and next-generation sequencing (NGS), and bioinformatic processing was performed as reported previously [38]. Produced variant calling files (VCFs) were processed with eVAI software (enGenome, Pavia, Italy; https://evai.engenome.com/#login; accessed on 13 June 2018) for annotation and variant prioritization, while the online VarSome tool (https://varsome.com/; accessed on 13 June 2018 [39]) was used to classify variants according to the American College of Medical Genetics and Genomics (ACMG) criteria (accessed on 1 November 2021). Gene copy number variations were analyzed with an inhouse pipeline. Patients were investigated for deletions or duplications using SALSA Multiplex Ligation-dependent Probe Amplification (MLPA) probemix P051-D2 and P051-D2 Parkinson’s disease (PD) assay and using P471-A1 EOFAD assay (MRC-Holland). The assays were performed according to the manufacturer’s recommendations, and data were analyzed using Coffalyser.Net software (MRC Holland, Amsterdam, The Netherlands).

#### 2.3.2. Hereditary Hypothesis Analysis

The hereditary hypothesis analysis was performed, using the son as proband, to highlight both variants “in common”, thus present in both mother and son that may have contributed to the common clinical features, and variants “not in common”, thus present only in the proband and probably linked to the clinical signs manifested only by the son. Firstly, variants were retained according to the following criteria: (i) variants with minor allele frequency < 0.01 on the ExAC, GnomAD, dbSNP, and ESP6500 databases; (ii) variants that fall in the gene coding regions, splice site regions, and variants that disrupt the 3′/5′-untranslated regions; and (iii) variants with a read depth > 15. Variants retained were subsequently filtered according to patterns of inheritance (Appendix A). Retained variants and relative genes were further investigated using different databases, such as OMIM (Online Mendelian Inheritance in Man; https://www.omim.org/; accessed on 14 June 2018), Decipher V9.29 (https://decipher.sanger.ac.uk/; accessed on 14 June 2018), HSF (Human Splicing Finder; http://www.umd.be/HSF/; accessed on 14 June 2018), MGI (Mouse Genome Informatics; http://www.informatics.jax.org/; accessed on 14 June 2018), STRING (https://string-db.org/; accessed on 14 June 2018), and GTEx (Genotype-Tissue Expression; https://gtexportal.org/home/; accessed on 14 June 2018) to assess their potential role in AD and LBD disease context. Final retained variants were validated using Sanger sequencing.

### 2.4. RNA Extraction and Whole-Transcriptome Sequencing

Frozen slices from four brain areas (PL, BG, HiC, and SN) of the two kindred cases and of the nondemented control were used to perform transcriptome analysis. Total RNA was isolated using Trizol® (Life Science Technologies, Leawood, KS, USA), and RNA quality and integrity were evaluated based on the RNA Integrity Number (RIN), acquired using the Agilent 2100 Bioanalyzer (Agilent RNA 6000 Nano Kit). RNA libraries were prepared in duplicate using the SENSE Total RNA-Seq Library Prep Kit (Lexogen, Vienna, Austria) according to the manufacturer’s protocol. Libraries were sequenced using NextSeq 500 Sequencer (Illumina, San Diego, CA, USA). FastQ files generation and transcript mapping and quantification were performed as described in Zucca et al. 2019 [40]. Differential expression analysis with respect to the healthy control was performed using R package EBSeq2 [41] and retained for further analysis with |log2(disease sample/healthy control)| ≥ 1, and an FDR ≤ 0.1. qPCR was performed to validate RNA-seq data. RNA dataset generated during the current study is available in the GEO repository (Reference number GSE193438).

### 2.5. Pathway and Gene Ontology Analysis

Gene set enrichment analysis was performed for the differentially expressed coding genes (DEGs). Gene Ontology (GO) enrichment analysis for biological processes, cellular components, and molecular functions and a KEGG (Kyoto Encyclopedia of Genes and Genomes https://www.genome.jp/kegg/; accessed on 5 May 2019) pathway analysis were conducted via enrichR web tool [42,43]. R software was further used to generate Dotplot graphs (using the ggplot2 library) and GO chord graphs using the Goplot library. GO semantics classes and pathways were considered significant for *p*-values < 0.05. Analyses were conducted by using all the DEGs (ALL_DEGs) and the separated upregulated (ALL_UP) and downregulated (ALL_DOWN) DEGs of each brain area of both mother and son. Only pathways and GO terms that were statistically significant were taken into consideration for further interpretations.

## 3. Results

### 3.1. Clinical Observation

Mother (BB105). The clinical history began at 78 years of age, with the development of a mild neurocognitive disorder (NCD) prevalent in memory and learning functions. Three years later, a progressive impairment of all cognitive domains leading to a diagnosis of dementia (major-NCD) arose. The patient died 12 years later, after typical, slowly progressive AD. Brain CT showed hippocampal and parietal atrophy. A cerebrospinal fluid (CSF) analysis was not performed. The clinical picture and course showed a slow but inexorable progression, scarcely influenced by drug therapies. At death, the CDR was 5 (terminal dementia).

Son (BB181). The son was a cultured and healthy man until the age of 68, when, a progressive cognitive impairment, especially in the memory and visuospatial areas, became evident. In less than one year, a condition of major-NCD was obvious. Brain MRI showed diffuse atrophy, and CSF analysis revealed a slight increase in total TAU (464 pg/mL) with phospho-TAU at the upper limit of the normal range (50 pg/mL) and Aβ peptide close to the lower limit (640 pg/mL). Due to memory and visuospatial deficits, his first diagnosis was probable AD with early posterior (occipital/parietal) involvement. The use of anticholinesterases resulted in a significant but transient improvement. Two years later, a frank parkinsonian syndrome with psychosis and fluctuations occurred raising the possibility of an LBD diagnosis. Due to the absence of relevant movement disorders and behavioral changes in the initial two years of the disease, a diagnosis of parkinsonism due to FTLD-TAU was ruled out. The patient was very sensitive to the action of neuroleptics; therefore, only quetiapine was administered to control the psychotic symptoms. The patient died of pneumonia one year later, only three years after the clinical onset. At death, the CDR was 3 (severe dementia). The final diagnosis was changed to probable mix dementia AD/LBD. Aside from his mother, he had no family history of dementia. His father died at a young age due to premature sudden cardiac death.

### 3.2. Neuropathology and Immunohistochemistry

Mother’s neuropathological picture confirms a definite AD diagnosis with high AD pathology, according to the ABC score (Thal stage V-A3; Braak stage VI-B3; and CERAD-C2). At the same time, it shows a mild to moderate presence of TDP-43 neuronal cytoplasmic inclusions in the amygdala and HiC and no α-syn deposits (Figure 1, mother’s brain images: A, D, G, and J showing BG, SN, PL, and HiC, respectively). Son’s neuropathological picture shows intermediate AD pathology, according to the ABC score (Thal stage III-A2; Braak stage III/IV-B2; and CERAD-C2). At the same time, it also shows a relevant LTS with severe LB and LN lesions in SN, HiC, temporal mesocortex, cingulate gyrus, and, to a lesser extent, parietal cortex, corresponding to a widespread brainstem–limbic–neocortical LTS (Beach’s stage IV). According to the McKeith scheme [10], such a neuropathological picture defines LBD/AD dementia mainly due to the presence of LTS and to a lesser degree to the presence of AD pathology. No TDP-43 inclusions are detectable (Figure 1, son’s brain images: B, E-E’, H, K, and L showing BG, SN, PL, and HiC, respectively). Moreover, both mother and son show moderate small vessel disease (SVD) with cerebral amyloid angiopathy. The brain pathology of the control case (BB118) is unremarkable, with mild to moderate SVD, mainly in deep white matter and basal ganglia, due to arteriolosclerosis. Amyloid deposits are detectable only in the neocortex (A1), and TAU pathology is limited to the entorhinal cortex (B1), corresponding to a picture of low AD pathology (Figure 1, control’s brain images: C, F, I, and M, showing BG, SN, PL, and HiC, respectively). 

### 3.3. Genetic Analysis

Focused exome gene panel was performed on the mother and son to screen the principal causative genes associated with AD, PD, and LBD. No pathogenic variants able to clearly explain the phenotype of the mother and the son were found. We further proceeded with the hereditary hypotheses analysis, merging the genetic data files of the two cases to create a common VCF with a total of 6734 genetic variants: 2461 present only in the mother, 2373 present only in the son, and 1900 present in both cases. Variants present in genes not associated with phenotypes similar to dementia or not involved in known pathways related to AD, PD, or LBD were discarded. Since no genetic information was available from the father, all variants present only in the son were considered de novo variants with respect to the mother genome. Hereditary hypotheses analysis highlighted six variants “in common” and six variants “not in common”, i.e., son-specific (Table 1). Among the “in common” variants, the c.12948_12950dupAAG in the Ryanodine Receptor Type-2 (RYR2) gene was the only variant defined as likely pathogenic by the ACMG guidelines. This variant leads to the insertion of an Arginine in position 4317, within the divergent region 1 that influences the sensitivity to Calcium (Ca^2+^) inactivation. Among the “not in common” (son-specific) variants, we found a heterozygous missense variant in the USP24 gene, classified as likely pathogenic according to the ACMG guidelines. Interestingly, USP24 falls in a locus (PARK10) significantly associated with PD [40]. Both patients were carriers of the heterozygous ApoE4 allele. MLPA and copy number variant (CNV) analyses did not reveal any relevant small or large genetic deletions/duplications in the two subjects (data not shown).

### 3.4. Transcriptome Profile

Whole-transcriptome analysis of four different brain areas (PL, BG, HiC, and SN) was performed to detect the most dysregulated brain area in the two patients and the associated pathways involved after data normalization on the nondemented control. The brain areas with the highest number of DEGs resulted in the SN for the son with 2733 DEGs, followed by the PL for the mother with 351 DEGs (Table 2). The HiC was the lowest dysregulated brain area, while the BG of the two cases were similarly dysregulated.

A principal component analysis (PCA) on all the DEGs derived from the two batches highlighted that the DEGs of the SN separate well for all the subjects from the DEGs of the other areas that instead overlap, except for the DEGs from one library replicate of the mother’s HiC, which was very different from all the other libraries and its replicate (Figure 2A). This difference may be due to the severe atrophy of the HiC of the mother. We then performed the PCA on each brain area separately (Figure 2B–E). For all the brain areas, replicates of each subject are consistent, and the replicates relative to each individual separate well with no overlaps among them. As expected by the number of DEGs, the PCA analysis revealed the SN of the son (Figure 2B) and the PL of the mother (Figure 2C) as the most different brain areas, by clear separation from the nondemented control. The PCA of the HiC did not reveal any substantial separation among the subjects (Figure 2D), while the PCA of the BG showed that both mother and son separate well from the control but are similar to each other (Figure 2E).

### 3.5. KEGG Pathway and GO Term Analysis

A KEGG pathway analysis for DEGs was performed according to Gagliardi et al. 2018 [44]. For each brain area of mother and son, the ALL_UP DEGs and the ALL_DOWN DEGs were considered separately to discover specific phenotype-related pathways as suggested by Hong et al. 2014 and Guo et al. 2019 [45,46]. A GO terms analysis for biological processes, molecular functions, and cellular components was performed for each brain area of both mother and son, distinguishing the ALL_UP DEGs from the ALL_DOWN DEGs. The most interesting data have been explained in this paragraph, starting from the two areas that showed a major deregulation in terms of gene expression, which are the SN area of the son and the PL of the mother.

#### 3.5.1. Substantia Nigra

**Son (BB181)** (Figure 3 and Figure 4). As expected by the high number of DEGs, the SN of the son presented the highest number of highly dysregulated pathways. Looking at the top 10 pathways related to the ALL UP DEGs, all reached statistical significance with the “Synaptic vesicle cycle” pathway being the most significant one (adjusted *p*-value < 0.0005) and more than 30 genes involved. Among the other pathways, we retrieved the “Nicotine addiction” pathway (adjusted *p*-value < 0.0005), which is related to the dopaminergic system, and the “Calcium signaling pathway” (adjusted *p*-value < 0.0025) (Figure 3A). Conversely, nine out of ten pathways related to the ALL DOWN DEGs reached statistical significance, with the “Hippo signaling pathway” being the most significant one (adjusted *p*-value < 0.02), together with pathways involved in cancer. Other pathways were instead related to the cardiac system, such as the “Dilated cardiomyopathy”, the “Arrhythmogenic right ventricular cardiomyopathy”, and the “Hypertrophic cardiomyopathy” pathways (adjusted *p*-value < 0.04) (Figure 3B). Concerning the GO term analysis, the top three GO biological process-enriched terms for ALL UP DEGs in the SN of the son were related to the chemical synaptic transmission, axonogenesis, and anterograde trans-synaptic signaling, which are also reflected in the top three GO cellular component terms related to the axon and the integral component of the plasma membrane. GO molecular function terms were related to cation (i.e., potassium) channel activity and syntaxin binding, involved in synaptic vesicles exocytosis. Among the top three GO cellular components and GO molecular functions, the enriched terms for ALL DOWN DEGs in the SN of the son were instead related to the integral component of the plasma membrane, the striated thin muscle filament, and the muscle alpha-actin binding, the last two being in line with the upregulated KEGG pathway involved in cardiac muscle contraction. GO biological process terms were instead related to pulmonary valve morphogenesis and development and to extracellular matrix organization.

The “Synaptic vesicle cycle” pathway, the most dysregulated pathway in the SN of the son, was further investigated (Figure 4). Transcripts related to the reuptake of the neurotransmitters and their loading into the synaptic vesicles were downregulated, while transcripts related to the synaptic vesicle fusion to the plasma membrane, such as SNAP25, synaptogamins (SYT1, SYT2, SYT3, SYT4, and SYT5), and the SNAP family (NSF, NAPA, NAPB, and NAPG) were all upregulated.

**Mother (BB105)** (Appendix A). GO term analysis of the ALL UP DEGs of the SN of the mother resulted in enriched terms related to the immune response, in accordance with the KEGG pathway analysis (Appendix A). Specifically, GO chord for biological processes and cellular components are related to neutrophils, tertiary granules, and MHC class II complex protein, while the GO chord for molecular functions is related to calcium and lipoprotein binding. Concerning the pathways related to the ALL DOWN DEGs, conversely to what was reported for the son, we found the “Synaptic vesicle cycle” together with the “GABAergic synapse” and the “Glutamatergic synapse” pathways, even if they did not reach statistical significance (Appendix A). The GO chord for biological processes and cellular components were related to the synaptic transmission, in particular, to the clathrin-coated vesicle exocytosis, linked to the downregulation of Synaptogamin 10 (SYT10) and Double C2 domain beta (DOC2B), involved in Ca^2+^-dependent intracellular vesicle trafficking transcripts.

#### 3.5.2. Parietal Lobe

**Son (BB181)** (Appendix A). Top ten KEGG pathways associated with the PL ALL UP DEGs of the son were related to “steroid biosynthesis”, “ferroptosis”, and “glycolysis/gluconeogenesis”. “Phagosome” and “synaptic cell cycle” pathways were also present (Appendix A). Associated GO-enriched terms for the biological processes, cellular components, and molecular functions were related to cholesterol metabolic process, response to metal ion, oxidoreductase activity, and endoplasmic reticulum lumen. Conversely, the top ten KEGG pathways related to the ALL DOWN DEGs were instead all related to cancer and viral infection (Appendix A). Accordingly, all the GO-enriched terms for the biological processes, cellular components, and molecular functions were focused on cytokine response, chemokine binding, and MHC protein complex (Appendix A).

**Mother (BB105)** (Figure 5). PL was the most dysregulated brain area of the mother. Among the top 10 KEGG pathways related to the ALL UP DEGs, we retrieved the “complement and coagulation cascade” pathway, the “longevity regulating pathway”, and the “prion disease” pathway (Figure 5A). However, none of these pathways reached statistical significance. Looking at the top 10 KEGG pathways related to the ALL DOWN DEGs, we found pathways related to lipid metabolism, such as the “steroid biosynthesis”, the “glycerolipid metabolism”, the “ribosome”, the “RNA transport” and the “protein processing in endoplasmic reticulum” pathways (Figure 5B). In addition, in this case, none of these pathways reached statistical significance. Concerning the GO term enrichment analysis, GO biological process-enriched terms concerning the ALL UP DEGs, were related to the response to reactive oxygen species and the glycosaminoglycan metabolism. The GO cellular component-enriched terms were related to the lysosome, while the GO molecular function-enriched terms were related to sodium symporter activity, protein homodimerization, and cadherin binding. Looking at the ALL DOWN DEGs, the GO biological process-enriched terms were related to renal cell differentiation and protein transport along microtubules; GO cellular components and GO molecular functions were related to nuclear inclusion body and ribonuclease P complex, linked to the downregulation of Ribonuclease P/MRP Subunit 1 (POP1), POP5, and ATXN3.

#### 3.5.3. Basal Ganglia

**Mother and Son** (Figure 6A). As predicted by the BG-specific PCA, mother and son were similarly dysregulated within this brain area with respect to the nondemented control. Particularly, regarding the top ten KEGG pathways associated with the ALL_DOWN_DEGs, up to six were found to be in common. In the same way, the GO biological process-enriched term “Chemical synaptic transmission” was found in both cases, associated with the DOC2A, GABRQ, GABBR2, and the SLC17A7 genes involved in neurotransmitter release (DOC2A), synaptic transmission inhibition (GABRQ, GABBR2), and glutamate reuptake into synaptic vesicles (SLC17A7). “Glutamate ionotropic receptor” and “Glutamate ionotropic binding” were also found among the top three GO-enriched terms related to cellular components and molecular functions of the mother and son, respectively.

#### 3.5.4. Hippocampus

**Mother and Son** (Figure 6B). Downregulated pathways in the HiC of the mother were linked mostly to cancer, while among the upregulated ones, there was “the mRNA surveillance” pathway as in the son, which was the only downregulated pathway retrieved together with the “Influenza A” pathway. None of the pathways found reached statistical significance in this brain area for both patients. These data are in line with the HiC-specific PCA; except for a duplicate of the mother, the two cases were similar to the nondemented healthy control subject (being closer with respect to the two principal component axes), predicted also by the small number of dysregulated transcripts found. GO-enriched term analysis for this area was not possible for the two cases due to the low number of dysregulated transcripts.

## 4. Discussion

In this paper, we characterize two kindred cases, mother and son, who were affected by a classic form of AD and by an aggressive form of dementia, firstly classified as probable AD and subsequently as an AD/LBD form. Patients were deeply characterized from a neuropathological, genetic, and transcriptomic point of view with the aim to highlight the possible reasons that triggered the son to develop such a different clinical dementia phenotype despite the common genetic background shared with the mother. Clinical diagnosis of both patients was confirmed by a thorough neuropathological examination. High AD pathology and widespread deposition of Aβ aggregates and diffuse TAU pathology were found in the mother’s brain, while a severe LTS was found in the son’s brain, accompanied by intermediate levels of Aβ aggregates and TAU pathology. Moreover, both mother and son showed moderate SVD with cerebral amyloid angiopathy.

Concerning their genetics, previous investigations showed that the two cases carried the ApoE ε4 allele, which is known to increase the risk of developing AD three to four-fold [47,48]. Accordingly, both showed a severe cortical amyloid load with amyloid angiopathy [49]. Interestingly, the ApoE ε4 allele has been also described as a potential genetic modifier of the age at onset, conferring anticipation through generations [50]. Regarding this specific case, both mother and son are carriers of the ApoE ε4 allele; therefore, the earlier age at onset of the son cannot be attributed to this genetic modifier, suggesting other underlying factors probably related to the different pathology shown by the two cases.

Through NGS, we found no pathogenic variants in the causative genes known to be involved in AD and LBD. Through the hereditary hypotheses, we found common variants that may have triggered dementia in both individuals and variants present only in the son that may be responsible for the parkinsonian spectrum and the LTS. Among the common variants present in both cases, the only one classified as pathogenic by the ACMG guidelines was a disruptive in-frame insertion (c.12948_12950dupAAG) in the RYR2 gene, which codifies for the Ryanodine Receptor Type 2 (RyR-2). RyR-2 is a Ca^2+^ channel expressed either in the sarcoplasmic reticulum of cardiac muscle and in the endoplasmic reticulum (ER) of brain neurons, including Purkinje cells in the cerebellum, and other neurons in the cerebral cortex and the dentate gyrus of the HiC [51,52]. Perturbed ER Ca^2+^ homeostasis is recognized as a central player in AD [53,54] and in axon terminals; indeed, ER Ca^2+^ release is involved in vesicle fusion and neurotransmitter release [55]. This evidence suggests that the mutation found in the two subjects may have altered the RyR-2 channel function in neurons, with consequences on the Ca^2+^ homeostasis during the aging trajectory of the brain, contributing to the synaptic loss and the development of dementia. Very recently, it has been demonstrated that increased RyR2 open probability induces neuronal hyperactivity and memory loss, two common manifestations of AD and AD progression [56]. Moreover, pharmacological treatment aimed to reduce RyR2 open time has been demonstrated to rescue AD-related deficits in mouse models, suggesting this molecular pathway as a potential target for addressing AD [57]. Among the “not in common” (son-specific) variants, we found a missense variant in the USP24 gene (PARK10), which encodes for a deubiquitinating enzyme, involved in protein turnover and degradation. It is well known that impairments of the ubiquitin proteasome system lead to a noneffective clearance of α-synuclein, with consequent accumulation throughout time and LB formation [58]. The variant in the USP24 gene may have impaired the deubiquitinating function, playing a role in the LTS of the son [59]. Moreover, USP24 has been previously associated with late-onset PD [60,61]. Although this association with PD remains to be elucidated, our results suggest that this variant may have influenced the pathology and the clinical phenotype of the son.

Through RNA-seq of four selected brain areas (PL, BG, HiC, and SN) of the two cases and the control case, we identified the most dysregulated brain areas of the two subjects, the SN in the son and the PL in the mother. Specifically, the SN of the son was found to be the brain area with the highest number of DEGs compared to all the other brain areas analyzed either in the subject and in the mother. Downregulated genes in this area were related to the synaptic signaling, mainly to the dopaminergic and the GABAergic synapses. Among the retrieved DEGs, TH, SLC6A3, and SLC18A2 were the genes with the largest differences in expression between the patient and the nondemented control, and all three were highly downregulated. TH codifies for tyrosine hydroxylase, which is involved in the conversion of tyrosine to dopamine in the dopaminergic neurons; SLC18A2 is involved in the loading of dopamine in the presynaptic vesicle for its release, and SLC6A3 codifies for the dopamine transporter (DAT), which is involved in the reuptake of dopamine in the presynaptic terminals. The downregulation of these three genes in the SN of the son may reflect the loss of dopaminergic neurons in the SN pars compacta, showing that the transcriptomic analysis strictly correlates with the neuropathological examination and may become a useful supportive tool in flanking challenging diagnoses. On the opposite side, among the most upregulated pathways in the son’s SN, we found the “Synaptic vesicle cycle” and the “Ca^2+^ signaling” pathways. While both pathways have been already reported in association with PD [62,63,64], no studies have linked these pathways to LBD yet. These data confirm that PD and LBD probably share the same pathogenetic pathways and represent two sides of the same coin. Transcriptomic analysis demonstrates a clear-cut difference between LDB in the son and AD in the mother in spite of the common clinical phenotype (dementia) and genetic background.

Unlike the son, the transcriptome analysis of the mother revealed that the PL was the most dysregulated brain area, according to the neuropathology, which showed an abundance of TAU deposition precisely in the PL. The parietal lobe is gaining even more attention in the development of AD [65,66,67]. Particularly, among the dysregulated pathways, we found the “steroid biosynthesis” and the “glycerolipid metabolism” ones. Several studies established a close relationship between alteration in steroidogenesis and fatty acid biosynthesis and neurodegeneration [68,69,70]. Moreover, pathways related to inflammation have been found dysregulated as expected in a late-stage AD brain, when inflammation contributes to neurodegeneration [71,72].

The other two brain areas, the HiC and the BG, were less dysregulated compared to the SN and the PL. Particularly, mother and son were similarly dysregulated within the BG according to their pathology, which was quite similar in this area and characterized by moderate amyloid deposition. It is noteworthy that in both cases the BG were not affected by specific proteinopathies (TAU/synuclein) but only by diffuse amyloid plaques, with little impact on the transcriptome. Conversely, the HiC was early affected by the disease and was an area with severe degeneration in both subjects, with double proteinopathy (TAU/synuclein) in the son. Nonetheless, this area appeared slightly dysregulated. Our data let us hypothesize that the amount of transcriptional dysregulation may be related to the level of damage accumulated across time by the different brain areas. The HiC, which was the first and most damaged brain area in both mother and son with the majority of dying or critically damaged cells, became less prone to respond to tissue damage through transcriptional dysregulation, reducing the transcriptional machinery’s ability to work at basal levels due to cell dysfunction or, possibly, with the aim to preserve at most cell survival. Brain areas with modest or late involvement of pathological changes, as in the case of the BG of the two cases, started to overcome tissue damage, resulting in a low and similar level of dysregulation. Lastly, areas affected by an evolving active pathology were highly and specifically dysregulated, as in the case of the SN and the PL in this study. They are still transcriptionally active and result in a peculiar transcriptional dysregulation depending upon the underlying pathogenesis and influencing the type of pathology. This work shows that the alterations of the transcriptome might follow the pathology according to a temporal and topographical trajectory where the less affected areas and, paradoxically, the most affected ones appear to be the least dysregulated, while areas with greater dysregulation are those in which the pathology is more active. Eventually, the interaction between genomics and transcriptomics may be crucial for identifying disease pathways and creating precise, powerful medications. Indeed, the identification of risk variants and disease-specific pathways, together with the possibility to follow brain changes throughout the disease by following the transcriptional dysregulation of different brain areas, might become helpful for the development of therapeutic strategies that target molecular pathways involved in AD and/or LBD, where effective treatments are still missing.

## 5. Conclusions

To our knowledge, this is the first study that compares the transcriptomic profile of different brain areas in kindred cases affected by different types of dementia, specifically AD and LBD. Here, we highlighted that the LBD pathology, compared to the AD, undergoes higher transcriptional dysregulation, especially within the SN, while in the AD context, a lower degree of transcriptional dysregulation was found, with the PL being the most involved brain area. In particular, we found a link between the level of transcriptional dysregulation in the different brain areas and the level of tissue damage accumulated across time. Even if AD and LBD may have a clinical overlap, our data highlight that the molecular bases of AD and LBD may follow very different pathways, and that transcriptome profiling through RNA sequencing may be an important tool in flanking the neuropathological examination for the differential diagnosis and for elucidating neuropathogenesis of AD and LBD.

## Figures and Tables

**Figure 1 biomedicines-10-01687-f001:**
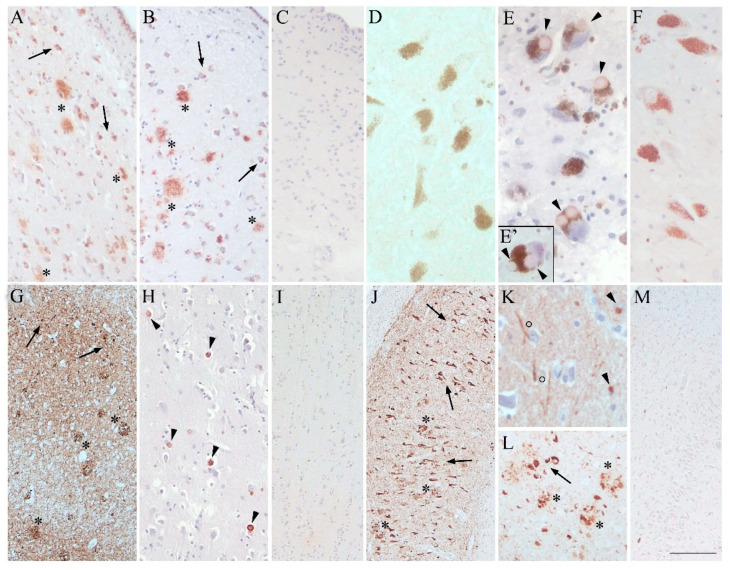
Comparison of the neuropathological picture among mother, son, and the control case. (**A**,**B**) the BG of mother and son show widespread diffuse and dense Aβ plaques (asterisks) and physiological intracellular amyloid in some neurons (arrows) (4G8 antibody; magnification 10×); (**C**) absence of Aβ in the BG of the control case (4G8 antibody; magnification 10×). (**D**–**F**) the SN of mother, son, and control, respectively (α-SYN-KM51 antibody; magnification 20×); only the son presents severe LTS with many LBs ((**E**); arrowheads); LBs are also easily detectable by using Hematoxylin &Eosin staining only (image (**E**’); magnification 20×). (**G**–**I**) the PL (cortex) of mother, son, and control, respectively; image G shows mother’s picture, characterized by the presence of strong pTau deposits with NFT (arrows) and NP (asterisks), according to her Braak stage VI (AT8 antibody; magnification 4×); image H demonstrates son’s picture, characterized by LTS with many LBs (arrowhead), according to his Beach’s stage IV (α-SYN-KM51 antibody; magnification 10×); image I shows the parietal cortex of the control case without any pathology (AT8 and α-SYN-KM51 antibodies; magnification 10×). (**J**–**M**) the HiC of the three cases; image (**J**) demonstrates mother’s picture with NFT (arrows) and NP (asterisks) due to severe pTAU pathology (AT8 antibody; magnification 4×); image (**K**) shows son’s neuropathology in the HiC, characterized by both LBs (arrowhead) and LNs (circles) (α-SYN-KM51 antibody; magnification 40×); image (**L**) proves the contemporary presence of pTAU lesions in the HiC of the son with NFT (arrows) and NP (asterisks) (AT8 antibody; magnification 10×); image (**M**) proves the absence of pathology in the HiC of the control case (AT8 and α-SYN-KM51 antibodies; magnification 4×). The 1 cm calibration bar in the lower right corner of the figure applies to all images corresponding to 85 µm in (**A**–**C**,**H**,**I**); 47 µm in (**D**,**E**,**E**’,**F**); 194 µm in (**G**,**J**,**M**); 35 µm in (**K**); and 107 µm in (**L**).

**Figure 2 biomedicines-10-01687-f002:**
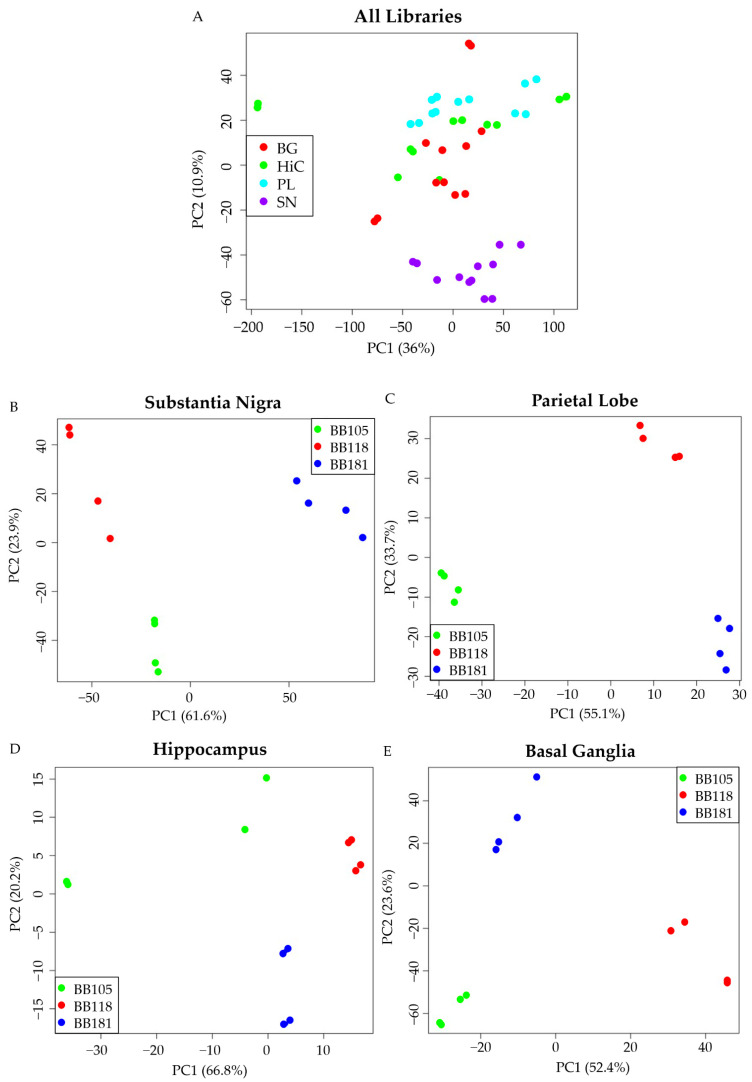
PCA of the RNA-seq libraries of each brain area in the two principal component spaces. (**A**) PCA of all the RNA-seq libraries together; (**B**) PCA of the substantia nigra; (**C**) PCA of the parietal lobe; (**D**) PCA of the hippocampus; (**E**) PCA of the basal ganglia. BB105: mother (AD); BB181: son (LBD); and BB118: nondemented control.

**Figure 3 biomedicines-10-01687-f003:**
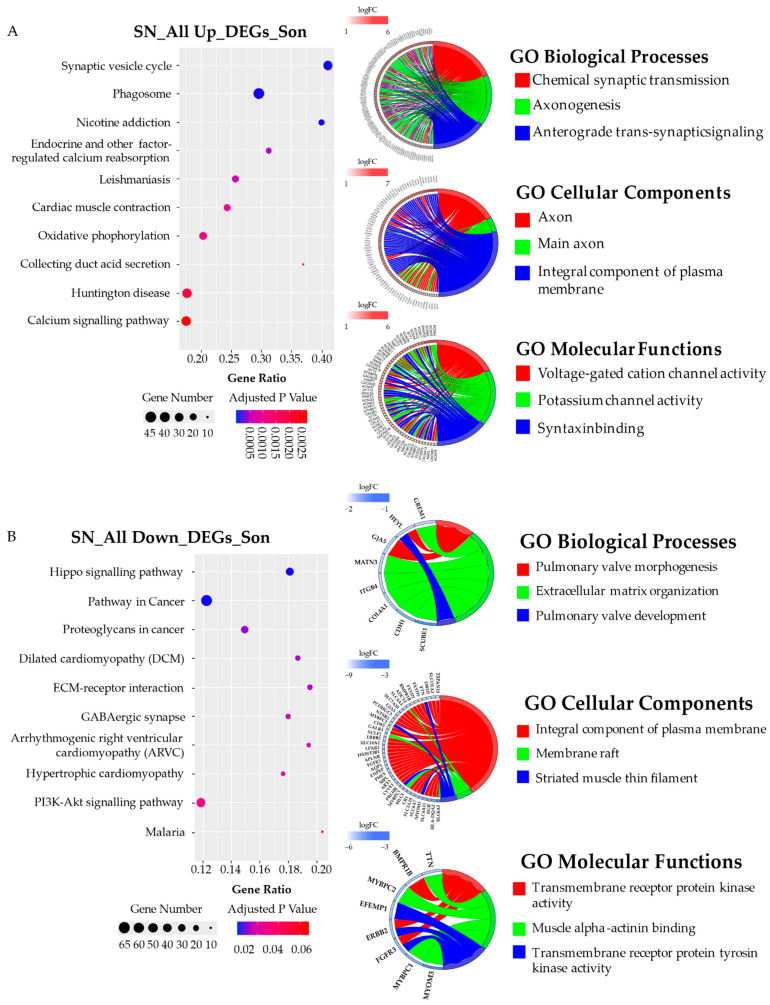
KEGG pathways and GO chord analysis relative to the DEGs of the substantia nigra of the son (BB181). (**A**) top ten KEGG pathways and top three GO-enriched terms in terms of biological processes, cellular components, and molecular functions related to the ALL UP DEGs; (**B**) top ten KEGG pathways and top three GO-enriched terms in terms of biological processes, cellular components, and molecular functions related to the ALL DOWN DEGs. The genes of the GOchords related to the SN_All Up_DEGs_Son have been listed in Appendix A.

**Figure 4 biomedicines-10-01687-f004:**
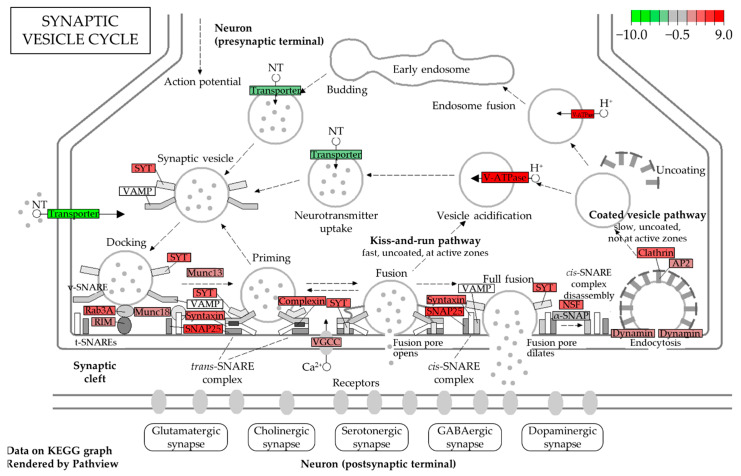
KEGG pathway related to the synaptic vesicle cycle. In green, the molecules associated with the downregulated DEGs, while in red the molecules associated with the upregulated DEGs found in the son [43].

**Figure 5 biomedicines-10-01687-f005:**
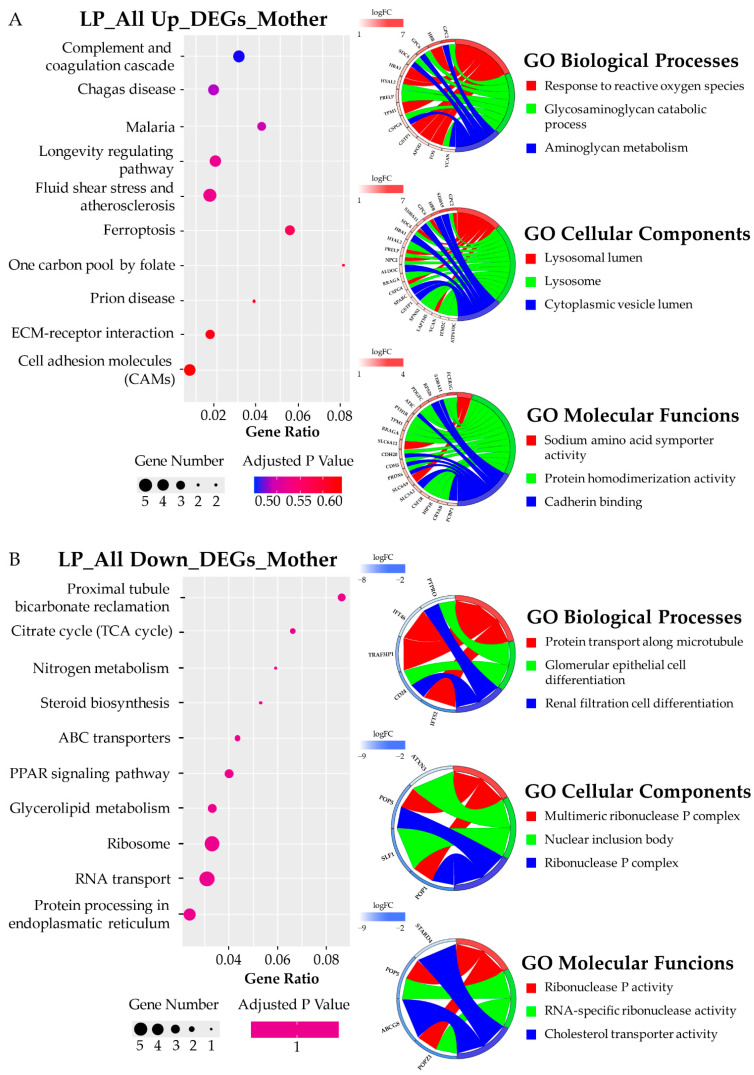
KEGG pathways and GO chord analysis relative to the DEGs of the parietal lobe of the mother (BB105). (**A**) top ten KEGG pathways and top three GO-enriched terms in terms of biological processes, cellular components, and molecular functions related to the ALL UP DEGs; (**B**) top ten KEGG pathways and top three GO-enriched terms in terms of biological processes, cellular components, and molecular functions related to the ALL DOWN DEGs.

**Figure 6 biomedicines-10-01687-f006:**
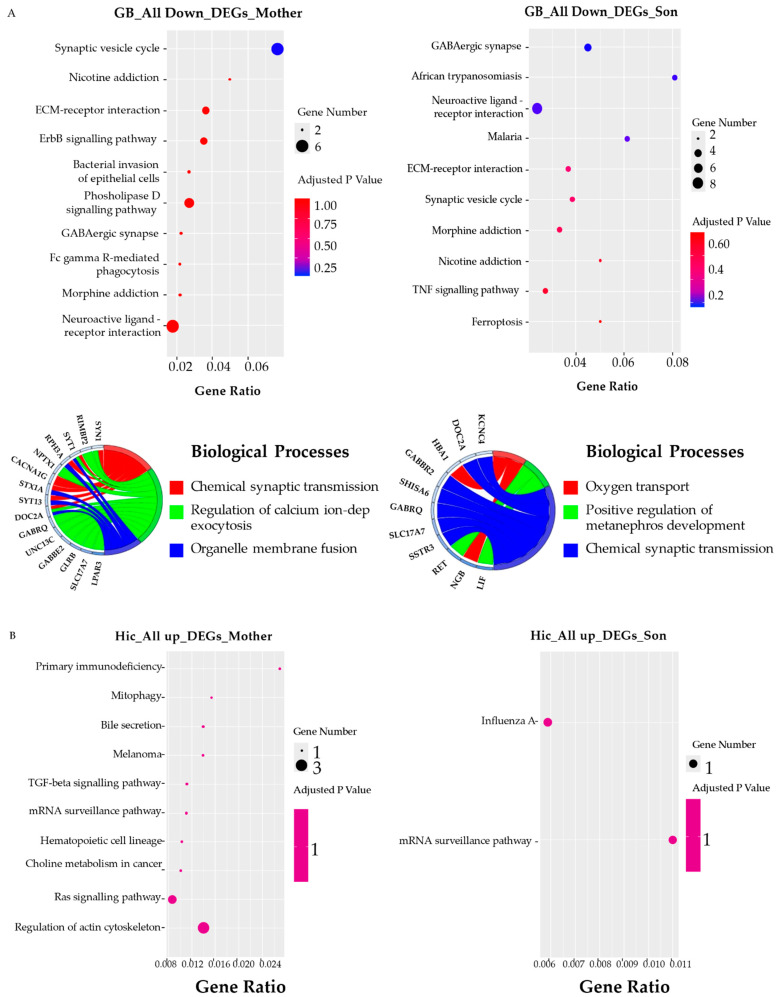
KEGG pathways and biological processes GO chords. (**A**) Top ten KEGG pathways and top three GO-enriched terms in terms of biological processes related to the ALL DOWN DEGs; (**B**) Top ten KEGG pathways related to the ALL UP DEGs.

**Table 1 biomedicines-10-01687-t001:** List of the variants found through hereditary hypothesis analysis. D: damaging; T: tolerated; NA: not available; LP: likely pathogenic; VUS: variant of uncertain significance; LB: likely benign; and B: benign.

	Gene	HGSV (Coding)	HGSV (Protein)	In Silico Prediction (SIFT; Polyphen2)	ACMG Classification
**Variants in common**	*RYR2*	c.12948_12950dupAAG	p.Gly4316_Ser4317insArg	NA; NA	LP
	*HTT*	c.1403-4T>A	p.??	NA; NA	VUS
	*CTNNA3*	c.1133G>T	p.Arg378Leu	D; D	VUS
	*CNTN6*	c.1585A>C	p.Ile529Leu	T; T	LB
	*SYNPO*	c.2507C>A	p.Thr836Asn	NA; T	LB
	*KCNT1*	c.522G>A	p.Met174Ile	T; T	B
	*HOMER2*	c.181A>G	p.Ile61Val	T; T	B
**Variants not in common (son-specific)**	*USP24*	c.2926G>A	p.Glu976Lys	T; T	LP
	*OBSCN*	c.184G>C	p.Gly62Arg	D; D	VUS
	*MTPAP*	c.-33A>G	p.??	NA; NA	VUS
	*GALC*	c.328 + 6A>G	p.??	NA; NA	VUS
	*DAPK1*	c.3287G>A	p.Ser1096Asn	T; T	LB
	*HDAC4*	c.2990T>C	p.Leu997Pro	T; T	LB

**Table 2 biomedicines-10-01687-t002:** Statistically significant differentially expressed coding RNAs in the four brain areas. In bold, the total number of DEGs of the most dysregulated brain areas of both mother and son. * log2FoldChange > 0; ** log2FoldChange < 0.

		n° DEGs Upregulated *	n° DEGs Downregulated **	Total n° of DEGs
**Substantia Nigra**	mother	181	116	297
	son	1567	1166	**2733**
**Parietal Lobe**	mother	156	195	**351**
	son	68	100	168
**Hippocampus**	mother	41	10	51
	son	5	1	6
**Basal Ganglia**	mother	103	159	262
	son	109	101	210

## Data Availability

The RNA-sequencing datasets presented in this study are publicly available in the Gene Expression Omnibus repository (https://www.ncbi.nlm.nih.gov/geo/; accession number: GSE193438; accessed on 11 January 2022).

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
