# Peer review of "Differential Neuropathology, Genetics, and Transcriptomics in Two Kindred Cases with Alzheimer’s Disease and Lewy Body Dementia"

_biomedicines, 2022, doi:10.3390/biomedicines10071687_

Round 1

Reviewer 1 Report

This paper presents the results of an extensive neuropathological, genetic and transcriptomic analysis of a mother-son pair both diagnosed with dementia. The authors present a critical evaluation of the data thus obtained and its implications, concluding that modern transcriptomic tools may lead to "a deeper understanding of AD and LBD pathogenesis".

Overall, I found this manuscript both innovative and well-presented. The following are areas which may benefit from clarification or correction from the authors:

1. The introductory discussion is brief and could benefit from some elaboration. As it stands, it consists of a single paragraph. It could be organized into 2-3 paragraphs, each with a specific focus: (a) prevalence and neuropathology of AD and LBD; (b) molecular mechanisms (common and divergent) in these conditions, including a discussion of other tau-related conditions linked to dementia (such as frontotemporal dementia), and (c) the state of the art in current genetic research, including summaries of single-gene and genome-wide studies examining the molecular genetics of AD, LBD and the divergences and overlaps between them.

2. The statement that LBD is the second most common form of dementia (lines 58-59) is open to question; research from lower-income countries suggests that vascular dementia is more common in these settings, and some authors have claimed a higher place for frontotemporal dementia (FTD) as well. A more balanced appraisal of the evidence, supported by a more recent study or meta-analysis (the existing citation is to a review article from 2007), can be provided here.

3. The second case presented in this paper raises interesting questions. The history is of cognitive deficits followed by parkinsonism and psychosis, but the MRI picture is of diffuse atrophy rather than the regional pattern seen in the mother. Therefore:

a) could this represent a case of an FTD variant ("FTD with parkinsonism"), also known to be associated with tau-related pathology?
b) were there any associated behavioural or personality changes in the initial two years of the illness?
c) was the patient receiving any additional medications (particularly antipsychotics) which could have caused parkinsonism?

4. In view of the familial nature of the cases being reported, it could be helpful to discuss the phenomenon of anticipation (i.e. reduction in age at onset in successive generations, perhaps due to a genetic cause) observed in some cases of this sort. See, for example, De Luca et al. (2019).

5. The ethical statement in this paper requires amendment. It is stated that informed consent was obtained from both patients participating in this study (lines 541-542). Was this obtained prior to a diagnosis of dementia, in the early stages of dementia, or from a relative / legally appointed representative in the later stages of the illness? This should be mentioned clearly in the paper.

6. Details of the selection of the control case require some elaboration. Was the subject from the same ethnic group / population as the two cases (to minimize differences simply due to inter-ethnic variations)? In view of the reports of an inverse correlation between Alzheimer's disease and some forms of cancer (see Zablocka et al., 2021) would it have been preferable to select a control case without a diagnosis of cancer (or of any other disease either positively or negatively correlated with Alzheimer's disease)? The authors should either provide a rationale for this (such as "liver cancer is not known to be associated with Alzheimer's disease" with supporting evidence) or mention it as a limitation of their study.

7. As this paper is being submitted for consideration in a journal with a focus on biomedicines, the authors should include some discussion of the potential therapeutic implications of their findings (for example, molecular "leads" towards novel treatment targets that could arise from larger transcriptomic analyses.)

Reviewer 2 Report

In this manuscript, Palmieri et al describe the neuropathology, genetics, and transcriptional alterations of a mother and son duo suffering from dementia, albeit of a different nature. The authors reveal a correlation between neuropathology and transcriptional changes in different brain areas, that allows for the clear separation of the two patients despite similar clinical manifestations. The mother, suffering from classic AD pathology, had high levels of A-beta and p-Tau deposition and greater levels of transcriptional changes in the parietal lobe. On the other hand, prominent involvement of the sustantia nigra with the presence of  a-synuclein-positive aggregates and widespread transcriptional changes were detected in the son's brain, in accordance with a PD/LBD diagnosis.

The data in this manuscript are well presented and informative, although the paper lacks any confirmatory experiments regarding the RNA-seq data. A minor comment is that the authors focused their bioinformatic analysis of the DEGs by separating them in two categories: all_up and all_down DEGs. It would be also interesting to analyze the data together as up and downregulated genes in the same pathway are possible and often found. 

In Figure 1, arrows or asterisks to indicate the presence of the relevant pathology would be helpful, as it would be to include images of the pathology in the control brain. 

Some of the acronyms used in the paper are not defined  (e.g., VCF, MLPA, CVN, and B in Table 1)

In the introduction, line 71 it should read "however, only a few limited information on the transcriptomes of the SN and the PL, which are key regions involved in LBD and AD neuropathology, are is present in the literature"

Round 2

Reviewer 3 Report

Please read attachment

Author Response

We would like to advise the reviewer that we also uploaded two figures in .tiff format of the α-synuclein staining in H&E and with the specific antibody, along with uploading the revised manuscript.
